# Small Molecule Inhibitors Targeting the Heat Shock Protein System of Human Obligate Protozoan Parasites

**DOI:** 10.3390/ijms20235930

**Published:** 2019-11-25

**Authors:** Tawanda Zininga, Addmore Shonhai

**Affiliations:** Department of Biochemistry, School of Mathematical and Natural Sciences, University of Venda, Thohoyandou 0950, South Africa; tzininga@gmail.com

**Keywords:** heat shock protein, Hsp90, Hsp70, inhibitors, obligate parasites

## Abstract

Obligate protozoan parasites of the kinetoplastids and apicomplexa infect human cells to complete their life cycles. Some of the members of these groups of parasites develop in at least two systems, the human host and the insect vector. Survival under the varied physiological conditions associated with the human host and in the arthropod vectors requires the parasites to modulate their metabolic complement in order to meet the prevailing conditions. One of the key features of these parasites essential for their survival and host infectivity is timely expression of various proteins. Even more importantly is the need to keep their proteome functional by maintaining its functional capabilities in the wake of physiological changes and host immune responses. For this reason, molecular chaperones (also called heat shock proteins)—whose role is to facilitate proteostasis—play an important role in the survival of these parasites. Heat shock protein 90 (Hsp90) and Hsp70 are prominent molecular chaperones that are generally induced in response to physiological stress. Both Hsp90 and Hsp70 members are functionally regulated by nucleotides. In addition, Hsp70 and Hsp90 cooperate to facilitate folding of some key proteins implicated in cellular development. In addition, Hsp90 and Hsp70 individually interact with other accessory proteins (co-chaperones) that regulate their functions. The dependency of these proteins on nucleotide for their chaperone function presents an Achille’s heel, as inhibitors that mimic ATP are amongst potential therapeutic agents targeting their function in obligate intracellular human parasites. Most of the promising small molecule inhibitors of parasitic heat shock proteins are either antibiotics or anticancer agents, whose repurposing against parasitic infections holds prospects. Both cancer cells and obligate human parasites depend upon a robust protein quality control system to ensure their survival, and hence, both employ a competent heat shock machinery to this end. Furthermore, some inhibitors that target chaperone and co-chaperone networks also offer promising prospects as antiparasitic agents. The current review highlights the progress made so far in design and application of small molecule inhibitors against obligate intracellular human parasites of the kinetoplastida and apicomplexan kingdoms.

## 1. Introduction

The most important obligate parasites that infect humans are mainly from the kinetoplastida and apicomplexa kingdoms. The kinetoplastid members represented by *Leishmania* species cause leishmaniasis, while *Trypanosoma cruzi* and *Trypanosoma brucei* cause “Chagas” disease and sleeping sickness, respectively. A member of the apicomplexan kingdom, *Toxoplasma gondii,* causes toxoplasmosis, and another prominent member of this kingdom, *Plasmodium falciparum,* causes the most severe form of malaria [1]. The life-cycles of most obligate human parasites traverse more than one host, and therefore, the parasites survive under distinct environmental and physiological conditions. Generally, the parasites are transmitted by arthropod vectors and the latter possess distinct physiological endowments compared to those of the human host. Some intracellular parasites such as *T. gondii* are transmitted by either ingestion of undercooked meat or ingestion of the oocyst passed in feces by infected cats. Irrespective of the mode of transmission, intracellular parasites need to adapt to physiological conditions prevailing in the host. The host immune response, which concomitantly is coupled to onset of fever conditions associated with clinical progression of infection, further places the parasites under stressful conditions. Apicomplexan parasites harbor an apicoplast, a plastid that is essential for survival at all stages [2]. The apicoplast is involved in the biogenesis of fatty acids and essential metabolic pathways. These functions define the essential role of the apicoplast in parasite survival [3,4]. Apart from its role in lipid biosynthesis, the apicoplast also hosts several nuclear encoded proteins such as DNA polymerase, DNA gyrase subunits, ribosomal proteins, molecular chaperones, and components of a Sulf-type Fe–S cluster assembly system [5]. For this reason, the translocation of nuclear encoded proteins into the apicoplast is important for parasite survival [6]. On the other hand, kinetoplastids are marked by the presence of a kinetoplastid, which is essentially a DNA-containing granule located within the mitochondria. It is constituted by a network of concatenated circular DNA molecules. Posttranscriptional editing of mitochondrial pre-mRNAs of kinetoplastids, as directed by small guide RNAs (gRNAs), produces functional mRNAs, giving rise to several metabolic proteins [7]. However, the kinetoplast is not essential for parasite survival within the mammalian host [8]. 

For parasites to survive, their proteome needs to be adept at meeting the demands of the hostile conditions prevailing within the alternating host/vectors environments, characterized by variable physiological conditions such as pH, temperature, and nutrient supply. Unlike free-living parasites, obligate parasites undergo extensive molecular evolution during their stint in the host [9]. This promotes production of mutated proteins, making their proteome generally aberrant [9]. It is for this reason that obligate parasites require a robust protein folding system for their survival in the host [10]. It has further been reported that 10% of the proteome of *P. falciparum* is characterized by prion-like motifs defined by glutamate-/asparagine-rich repeats [11]. Thus, nearly 10% of the *P. falciparum* proteome is prone to misfolding. The development of intracellular parasites at the various growth stages is characterized by distinct proteomic constituents. For example, it has been suggested that a third of the *T. brucei* proteome undergoes at least a two-fold difference in abundance for parasites cultured in blood stream form (BSF), representing the mammalian host stage versus parasites cultured at the procyclic form (PCF), the latter representing conditions that prevail within the insect vector [12,13,14]. This suggests that the protein folding machinery of obligate parasites needs to be adaptable to the demands of a life characterized with dynamic physiological changes. Notably, members of the kinetoplastid family harbor an expanded heat shock protein machinery as a mechanism for survival within the host [15].

Heat shock proteins (Hsps) form part of the cell’s response to stress. Hsps are conserved and ubiquitous molecules. Hsps are molecular chaperones that facilitate folding of other proteins (substrates/clients) in the cells, and hence play a particularly important role during cellular stress. Hsps are also implicated in translocation of proteins and protein complex formation. The latter role is important, especially in promoting host–parasite interaction. In this respect, heat shock proteins modulate adhesion of surface proteins of the parasite to the host cell to facilitate efficient host cell invasion [16,17]. In support of this, it has been reported that Hsp20 is important for the motility and infectivity of sporozoites of the malarial agents, *Plasmodium vivax* [18]. 

## 2. Hsp90 Family 

Heat shock protein 90 (Hsp90) is an ATP-dependent molecular chaperone that play a central role in a variety of cellular processes, including cell cycle control, cyto-protection, and cell signaling pathways [19]. Hsp90 is a ubiquitous protein that is found in eubacteria and eukaryotes. Structurally, Hsp90 is composed of an N-terminal domain (NTD) that serves as the ATP binding site, which in turn is linked to the middle domain (MD) through a charged linker segment [20]. The MD is adjacent to the C-terminal domain (CTD), which, in the case of cytosolic Hsp90 isoforms, is defined by MEEVD residues [21].

### 2.1. Hsp90–Hsp70 Chaperone Pathway

Hsp90 and another molecular chaperone, Hsp70, cooperate to fold select proteins in the cell, such as kinases, transcription factors, and steroid hormone receptors, most of which are implicated in cellular development [22]. In addition, Hsp90 and Hsp70 occur in functional networks with other chaperones and co-chaperones (molecules that regulate their function). The functional interaction between Hsp70 and Hsp90 is modulated by Hsp70–Hsp90 organizing protein (Hop/Sti1; [23]). Hop serves as an adaptor protein linking Hsp90 and Hsp70, thus allowing them to exchange substrates (Figure 1) [24]. Both cytosolic Hsp90 and Hsp70 possess C-terminal MEEVD (MEQVD in *Leishmania major* Hsp90) and EEVD motifs, respectively, that facilitate interaction of the chaperones with Hop [25,26,27]. On the other hand, Hop binds to Hsp90/Hsp70 through its multiple tetratrico-peptide repeat domains (TPR1, TPR2A, TPR2B) [27,28]. 

A co-chaperone of Hsp70, Hsp40, is responsible for recruiting substrates to Hsp70 and also stimulates ATP hydrolysis by Hsp70 [29,30]. The ADP-bound state of Hsp70 possesses high affinity for the substrate. A nucleotide exchange factor then facilitates release of ADP to make way for ATP, resulting in the release of the fully or partially folded substrate. Substrates that require Hsp90 for complete folding are handed over to Hsp90 for further refolding. In this regard, Hsp70 binds first to the TPR1 domain of Hop and the resultant conformational changes make the TPR2A domain of Hop accessible to Hsp90 [24]. Subsequently, Hsp70 shifts from the TPR1 to the TPR2B domain, and the latter event is linked to substrate transfer from Hsp70 to Hsp90 [24]. 

### 2.2. Hsp90 as a Drug Target

Several factors make Hsp90 amenable to drug targeting. It is ubiquitously expressed and essential for the survival of *P. falciparum*, *T. gondii,* and *Leishmania* [31,32,33]. Various inhibitors of Hsp90 target its domains NTD; MD; CTD; and the C-terminal EEVD motif) (Figure 2) [34]. Since the NTD of Hsp90 is responsible for its ATPase activity, ATP mimics and any compounds that inhibit the enzymatic activity of the molecular chaperone constitute some of the most effective inhibitors [35,36]. For example, human Hsp90 exhibits low basal ATPase activity, and on the other hand, Hsp90s of parasitic origin exhibit fairly high basal ATPase activities [37]. This suggests that despite its apparent sequence conservation, Hsp90 possess distinct functional features across species, an aspect that could be exploited in the targeted inhibition of the parasite protein. It has been further reported that human or *P. falciparum* Hsp90 confers differential inhibitor sensitivities to yeast cells in a complementation assay [38]. The high basal ATPase activity of parasite Hsp90, such as the *P. falciparum* cytosolic homologue, PfHsp90, is more sensitive to inhibition than its human homologue [39]. This suggests that inhibition of Hsp90 is more detrimental to the parasite than it is to humans. The activities of both Hsp70 and Hsp90 are modulated by various co-chaperones and the trade-off from this is amplification of the functional versatility of the two chaperones [40]. For example, an investigation into the effects of 12 co-chaperones of Hsp90 concluded that co-chaperones of Hsp90 determine the fates of the latter’s battery of client substrates [41]. A previous study further noted that none of the common co-chaperones of Hsp90 were present in more than one organism out of a pool of nine disparate organisms investigated [42]. This suggests that the variable distribution of co-chaperones of Hsp90, and indeed those of Hsp70 [43,44], within human and parasite systems could present a chink in the armor of the parasite with respect to selectable inhibition. 

Some of the inhibitors of Hsp90 are synthetic peptides that specifically bind to its C-terminal EEVD motif [45,46]. These are in the form of TPR-based mimetics, which competitively bind to the Hsp90 EEVD motif, inhibiting interaction with TPR co-chaperones. This may be viewed as an indirect inhibition mechanism for abrogating downstream metabolic pathways which depend on Hsp90 clients for their function. Peptide-based drugs derived from amphibians have shown promising indices against members of *Plasmodium* and *Leishmania* [47]. However, the shortcomings of peptide-based chemotherapeutics are high production costs in comparison to conventional organic molecules, short half-lives, poor proteolytic stability, and low lipid bilayer penetration [48]. Their limited cell penetration capabilities would require application of higher dose (in micromolar) ranges versus the nanomolar ranges for organic molecules [47]. In addition, the poor bioavailability of peptide-based antiplasmodial agents has dampened their prospects as drug candidates [49,50]. However, efforts towards identifying effective peptide-based antiparasitic agents need to be further conducted.

### 2.3. Small Molecule Inhibitors of Hsp90 

Most of the Hsp90 inhibitors identified so far compete with ATP for binding onto the NTD of the protein [51]. These inhibitors trap Hsp90 in an inhibitor-bound state, leading to abrogation of the chaperone’s function, and the net effect is that maturation of Hsp90 client proteins important for cell survival such as kinases, phosphatases, and transcription factors is aborted [52]. So far, several natural and synthetic drugs targeting Hsp90, including natural ansamycin and derivatives of purine, resorcinol, benzamide, aminopyri(mi)dines, and tricyclic imidapyridines, have been described (Figure 2) [53,54,55]. Independent studies have evaluated the antiparasitic activity of some anticancer Hsp90 inhibitors (Figure 2), including ansamycin (GA, 17-AAG, 17-DMAG), benzamides (SNX-5422, SNX-2112), resorcinol (onalespib, luminespib, ganetesbip [STA-9090]), and a purine scaffold BIIB021, PU-H71 [56,57]. However, the main shortcoming of these anticancer agents as antiparasitic agents has been lack of selectivity between host and parasite Hsp90, resulting in hepatoxicity [58,59,60]. 

Hsp90 cooperates with several co-chaperones, and thus, one effective way to disrupt its function is to abrogate its interaction with functional partners. However, research on this aspect is only at the formative stages. Small molecule mimetics of TPR, such as Antp-TPR [61], have been described, but their possible role as antiparasitic agents is yet to be explored.

### 2.4. Plasmodial Hsp90 as a Drug Target

Four Hsp90 paralogs—cytosolic Hsp90 (PF3D7_0708400), endoplasmic reticulum Grp94 (PF3D7_1222300), mitochondrial tumor necrosis factor type 1 receptor-associated protein (TRAP1; PF3D7_1118200), and an apicoplast Hsp90 (PF3D7_1443900) (Figure 3)—are represented on the *P. falciparum* genome [65]. There have been recent efforts to repurpose anticancer Hsp90 inhibitors towards addressing several parasitic diseases, including *P. falciparum* malaria. Among these potentially repurposed compounds, 17-DMAG and NVP-AUY922 (luminespib) have been reported to exhibit high affinity for plasmodial Hsp90s (cytosolic PfHsp90 and ER Grp94) [55,56]. In addition, 17-DMAG and luminespib have been shown to exhibit potent cytotoxicity to chloroquine-resistant *P. falciparum* parasite growth [56]. However, using cell culture assays, 17-DMAG and luminespib were reported to be incapable of selectivity for parasite Hsp90 without affecting human Hsp90 [55,56,60]. This may be attributed to the functional conservation of Hsp90 across species. 

Despite the high conservation of Hsp90 “Bergerat fold” ATPase domain [66] and the respective ATP binding motifs, as illustrated by the multiple sequence alignments (Figure 3), prospects exist for targeting parasite Hsp90 with minimum effects on the human system [67]. For example, a study by Wang and colleagues [67] identified aminoalcohol-carbazoles that exhibited much higher affinity for PfHsp90 than for the human orthologue.

Crystallization of Hsp90 NTD complexed to ADP [68] and its inhibitors GA [69], radicicol [35,70], would serve as a platform for developing compounds targeting this domain of the protein [71]. However, there has been limited success with respect to the development of compounds targeting the NTD of Hsp90. In silico approaches were used to design a purine derivate, PU-H17 (Figure 2) [72], as an inhibitor of Hsp90. PU-H17 worked in synergy with chloroquine to improve survival of mice against malaria (Figure 3) [60]. However, PU-H17 was reported to exhibit higher affinity for human Hsp90 than PfHsp90 [55]. This limited its clinical prospects as a possible antimalarial monotherapy. Despite the limited success in the development of Hsp90 inhibitors, Wang and colleagues [67,73] designed potent PfHsp90 inhibitors, 7-azandole derivative (IND31119), and an aminoalcohol-carbazole (N-CBZ) compound (Figure 2). These compounds target a unique hydrophobic pocket which extends from the base of the conserved Hsp90 ATP binding pocket. This unique motif distinguishes PfHsp90 from the rest of the Bergerat fold-based ATPases and presents a new avenue for developing Hsp90-specific inhibitors. In addition, utilizing the same rationale, there have been attempts to design Hsp90 inhibitors with ansamycin, azalomycin backbones against malaria [74].

In light of persistent resistance against most monotherapeutic interventions against malaria, efforts have shifted to combination therapy, as currently recommended by the World Health Organization. Targeting of Hsp90 in combination therapy has been shown to be effective in reversing resistance to chloroquine by *P. falciparum* [60]. In addition, two harmine-based inhibitors of PfHsp90 (harmine 17A and harmine 21A) were reportedly effective at reversing both chloroquine and artemisinin resistance based on a mouse malaria model (Figure 2) [60,64,75]. This further highlights the prospects of targeting heat shock proteins in combination therapies against parasites to recover the efficacy of traditional drugs to which parasites are currently resistant. 

### 2.5. Leishmanial Hsp90 as a Drug Target

*Leishmania* Hsp90 (LdHsp90) is essential for the proliferation and survival of the parasite at the amastigote stage. Treatment of *L. donovani* with GA arrests parasite growth, inhibiting its transformation from promastigote to the amastigote stage [76]. In addition, the absence of a key Hsp90 co-chaperone, p23, in *Leishmania* is thought to enhance sensitivity of the parasites to both GA and radicicol [77,78]. This suggests that co-chaperones of Hsp90 may augment parasite drug resistance, and such a phenomenon would further justify the importance of combinational therapies against parasitic infections. *Leishmania braziliensis* Hsp90 (LbHsp90) was shown to be selectively targeted by alkynyl substituents of reblastatin (Figure 2) obtained from *Streptomyces hygroscopicus* (GA producer) [79]. Thus, identification of species-specific Hsp90 inhibitors remains a promising agenda.

Most of the ansamycin-based Hsp90 inhibitors have failed to progress beyond clinical trials against cancer due to their low solubility [74,80]. To overcome this, 17-AAG was combined with hydroxypropyl-β-cyclodextrin (HPβCD) to improve its solubility [81]. The resultant complex was effective at killing *Leishmainia amazonensis* at IC_50_ of 0.006 nM [81]. This further raises prospects for the development of potent anti-Hsp90 inhibitors against intracellular parasites. 

### 2.6. Toxoplasma Hsp90 as a Drug Target

*T. gondii* Hsp90 (TgHsp90) is an essential molecule, that plays an important role during the transition from tachyzoite to bradyzoite stages [82]. As a consequence, the application of GA to inhibit TgHsp90 resulted in growth stage arrest of the parasite [82,83]. In addition, treatment of the parasite using GA compromised its virulence [82]. Despite TgHsp90 being an attractive drug target, the unavailability of a three-dimensional image of the protein hampers the design of novel inhibitors targeting this protein. 

### 2.7. Trypanosoma Hsp90 as Drug Targets

*Trypanosoma brucei* Hsp90 (TbHsp90) has been studied as a drug target and its inhibition achieved nanomolar potency with GA, radicicol, and luminespib [84]. In addition, GA derivatives exhibited high selectivity for TbHsp90. Similarly, treatment of *T. cruzi* with GA abrogates TcHsp90 function [85]. Similar findings were observed from further studies on trypanosomes using the classical ansamycin GA derivatives (17-AAG and 17-DMAG) against both *T. brucei* [57] and *Trypanosoma evans* [37]. Treatment of *T. brucei* parasites with 17-AAG exposed the parasites to heat shock [84]. In addition, a derivative of the anticancer agent, resinol, luminespib, and a purine scaffold, CUDC-305, abrogated Hsp90 function (Figure 2) [57]. Furthermore, CUDC-305 exhibited access to the blood–brain barrier and this is important for its efficacy at late stages of *T. brucei* infection [86]. Using biophysical and biochemical techniques, Pizarro and colleagues [87] identified three short regions representing sequence divergence amongst the Hsp90 NTDs that they targeted for selective inhibition between human Hsp90 and the *T. brucei* homologue using compounds **1**, **3**, and **4** (Figure 2). Compound **5**, a benzamide derivative, was found to be selective against TbHsp90, and, for this reason, it is a promising lead compound [87]. However, despite the promising therapeutic indices of these prospective drugs, there is need to improve their safety and efficacy.

## 3. Hsp100 Family

The Hsp100 family of molecular chaperones is largely involved in the untangling of protein aggregates and complexes of polypeptides [88]. The general structure of Hsp100 is represented by that of ClpB (caseinolytic protease B) of prokaryotic origin and the Hsp104 homologue of eukaryotic origin (absent in mammals), both of which possess two nucleotide binding domains (NBD1 and NBD2; AAA^+^ domains) (Figure 4) [89]. The two subdomains are important for the formation of functional Hsp100 hexameric complexes (Figure 4). Hsp100 possess ATPase activity, which is thought to be modulated by substrate binding [90,91].

The N-terminus of Hsp100 serves as its substrate binding domain [92]. The functions of Hsp100 as both a dis-aggregase and ATPase are modulated by a conserved residue, D184, located in the NBD1 of Hsp104 [90]. Residue D184 is analogous to residue D178, present in ClpB [90]. Residue D184 serves as a modulator of both the ATPase and dis-aggregase functions of Hsp100 [89]. Hsp100 cooperates with Hsp70 to form a dis-aggregase complex that threads polypeptides from their aggregates [89,93]. It is postulated that Hsp100 binds to aggregated substrates first, and threads them out from the mesh, to ultimately expose their hydrophobic patches. The exposed hydrophobic patches are recognized by Hsp70, which then bind to the misfolded proteins towards refolding them [89,94,95]. The rest of the Hsp100 family (ClpC and ClpA) share the same domain architecture with Hsp104 and ClpB (Figure 4) [96]. ClpC and ClpA form functional complexes with ClpP to facilitate its role in protein degradation [96].

### Parasitic Hsp100 as Drug Targets

*P. falciparum* encodes eight Clp proteins. Of these, five members belong to the ClpATPases [AAA+ superfamily (PfClpB1, PfClpB2, PfClpC, PfClpM, PfClpY)] [97,98] and three to proteases-like proteins (PfClpP, PfClpR, and PfClpQ) [97,99]. Three members of the Clp AAA+ proteins are localized in the apicoplast, while PfClpB2 occurs in parasitophorous vacuole (PV) and PfClpY is resident in the mitochondrium [97]. ClpB has been implicated in the secretion of malarial proteins into the PV [98,100]. Clp proteins form heptameric functional complexes that play an essential role in the development of the apicoplast [98]. 

The so-called *Plasmodium* translocon of exported proteins (PTEX; Figure 5) [100], which is essential for the trafficking of parasite proteins into the host cell, is comprised of a *P. falciparum* Hsp101 (PfClpB2) [101,102]. PfHsp101 plays an important role in receiving protein cargo dispatched by the parasite for export to the infected red blood cell (iRBC), and the chaperone further threads the protein cargo to facilitate its trafficking via the PTEX towards the parasite-infected RBC [102,103,104]. This threading process is undertaken by six NBD2 loops of PfHsp101 that cyclically channel unfolded substrate through a spiral platform [102]. The resolved structures of the PTEX translocon from *P. falciparum* while in action, threading substrates meant for export to the iRBC, provide the basis for the design of novel inhibitors of this essential pathway [102]. Encouragingly, the human system does not possess a PTEX equivalent, hence this makes this pathway selectable against malaria parasites [100,105]. Attempts to use the p97-based ATP competitive inhibitor, isoquinolone, N2, N4-dibenzylquinazoline-2,4-diamine DBeQ [106] to inhibit the PTEX demonstrated strong antiplasmodial activity [107]. It is interesting to further note that the PTEX translocon does not occur in other members of the apicomplexan group apart from plasmodial species [100]. TgClpB1 and TgClpB2 represent cytosol and mitochondrial forms of Hsp100 and are known to be essential for parasite growth [108]. This makes them potential drug targets. However, in *L. major,* the ClpB proteins are dispensable for normal growth, although they are important for thermotolerance [109]. Taken together, this suggests that the Hsp100 family represents promising druggable candidates against both kinetoplastids and apicomplexan species, as some of their isoforms only occur in parasites but are not represented in the human system. 

## 4. Hsp60 Family

The Hsp60 family from prokaryotes functions as a complex composed of two essential proteins: GroEL (chaperonin 60) and GroES (chaperonin 10) [112]. In eukaryotes, Hsp60s are localized to the mitochondria and chloroplast in plants [113]. On the other hand, the Hsp60 homology t-complex polypeptide-1 (TCP-1 complex) is localized in the cytosol of eukaryotic cells [114,115]. TCP-1 is comprised of a heterocomplex, which facilitates the assembly of actin and tubulin [114]. TCP-1 is important in folding of nascent client polypeptides from the ribosomes [114]. On the other hand, the prokaryote-based GroEL/ES system is involved in the assembly of multi-protein complexes by forming a central cavity into which substrates are sequestered for folding [116,117]. The central cavity forms an isolation chamber and GroES constitutes the lid which closes the GroEL cavity [118]. The GroEL/ES complex is a double ring composed of 14 subunits [116]. Each subunit is subdivided into three domains (apical, intermediate, and equatorial). The apical domain forms part of the central cavity that directly binds to substrates through hydrophobic interactions [116]. The apical domain binds to the substrate, while the equatorial domain binds and hydrolyses ATP [119]. The intermediate subdomain acts as a hinge located between the apical and equatorial domains, and facilitates allosteric transmission of signals induced by ATP [119,120]. The conformational changes on the substrate binding surface dictate the substrate affinity [121]. Hsp60s binds to substrate when the chaperonin is in hydrophobic state [122]. Thus, the first ATP binding event results in higher affinity status (enhanced hydrophobicity) and binding of the subsequent ATP results in lower affinity state (attainment of hydrophilic state), leading to substrate release [119,123]. 

### Targeting Parasite Hsp60 

Most of the compounds known to inhibit parasite Hsp60/10 chaperone systems have low selectivity, as they inhibit *Escherichia coli* GroEL/ES and human Hsp60/10 in vitro [124]. Suramin, a polysulphonated symmetrical naphthalene derivative and a potent inhibitor of the Hsp60/10 chaperone system, is used as first line chemotherapeutic agent against *T. brucei gambiense* and *T. brucei rhodesiense*), an agent for African sleeping sickness (Figure 6) [125]. Although suramin has been shown to be selective for parasitic Hsp60 [126], its effectiveness in cell-based systems has been limited due to poor access of the drug to the chaperonin, which is remotely located in the mitochondrial matrix [127]. The *T. brucei* genome encodes for three Hsp60s and it has not been confirmed which one is targeted by sumarin [128]. It should further be noted that suramin’s definite mechanism of action is not known, although it has been shown to inhibit several glycolytic enzymes [129,130]. *T. brucei* Hsp60/10 inhibitors include closantel and rafoxanide, both of which inhibit *E. coli* and human Hsp60 [130,131]. In addition, epolactaene and myrtucommulone have been proposed as inhibitors of Hsp60/10 in cancer cells (Figure 6) [132,133]. Another promising anticancer drug, KHS101, selectively targets glioblastoma cells, which makes it a useful lead towards further development of inhibitors of Hsp60s of parasitic origin [134]. Known Hsp60 inhibitors abrogate the refolding activity of the chaperonin without affecting its ATPase activity [124]. However, their broad mechanism of action is still to be defined.

The *P. falciparum* genome encodes for two GroEL chaperonins: PF3D7_123100 (PfCpn60) that localize in the apicoplast, and PF3D7_1015600 (PfHsp60) that localize to mitochondria [135]. The *P. falciparum* genome also encodes for two GroES homologues: PF3D7_1333000 (PfCpn20) resident in the apicoplast, and PF3D7_1215300 (PfCpn10), which is resident in the mitochondria [134]. At the blood stage, *P. falciparum* survives inside the RBC and utilizes hemoglobin as a source of nitrogen. For this reason, the parasite breaks down hemoglobin to salvage nitrogen [136]. The hemoglobin degradation system comprises a complex formed between Hsp60 and calpain, a cysteine protease [136]. Thus, the PfHsp60–PfCalpain partnership may constitute a druggable target against the parasite. An antitumor agent, mizoribine (imidazole nucleoside antibiotics; Figure 6), is known to target Hsp60 [137] and was previously shown to inhibit parasite growth, possibly through disruption of the PfHsp60–PfClapin pathway [136]. 

*T. gondii* expresses two closely related Hsp60s, which are products of alternative splicing [138]. TgHsp60 is essential for parasite survival and is involved in regulating the respiratory pathway of the parasite and its various development stages [138]. Inhibitors targeting *T. gondii* Hsp60 were identified using in silico modeling [139]. Using the same approach, a small molecule inhibitor that attaches to the ATP binding domain of TgHsp60 was identified, although experimental evidence for this is pending [139].

## 5. Hsp70 Chaperone System

Hsp70 superfamily is one of the most conserved Hsps, whose members are represented in all domains of life [140]. In archaea and eubacteria, Hsp70 is generally referred to as DnaK. Hsp70s are ubiquitous molecules that play important roles in protein quality control, and some of their homologues are stress-inducible. The various localization profiles of Hsp70 facilitate specialized function. The Hsp70 domain architecture is generally comprised of an N-terminal, ~44 kDa nucleotide binding domain (NBD) (Hsp70_NBD_), which hydrolyzes ATP (Figure 7). On their C-terminus is located a substrate binding domain (SBD) (Hsp70_SBD_) and lid subdomain. The Hsp70_NBD_ is more conserved than the Hsp70_SBD_, suggesting that the latter confers specialized functional features to the protein [141]. The two domains are connected by a highly conserved 7-residue linker segment (Figure 7) [142]. Hsp70 is divided into two sub-families: DnaK-like (canonical Hsp70s) and Hsp110 [141,143]. DnaK/canonical Hsp70 is capable of refolding misfolded proteins and suppressing protein aggregation [144].

Hsp70 exhibits low basal ATPase activity in which one molecule of ATP is hydrolyzed by one Hsp70 protein in ~25 min at 30 °C [145]. Hsp70 possesses lower affinity for substrate in its ATP-bound state, and this situation facilitates the release of substrate [146,147]. However, when Hsp70 is complexed to ATP, substrate, and Hsp40, its ATPase activity is raised by 1000-fold [148]. On the other hand, Hsp70 exhibits higher substrate affinity in the ADP-bound state, and hence, in this state, it retains the substrate for much longer than when it is bound to ATP (Figure 1; [147]). Thus, nucleotides regulate the substrate bind and release cycles of Hsp70. Hsp70 is an ATP-powered protein refolding machine. Although Hsp70′s basal ATPase activity is generally low, it cooperates with its co-chaperone, Hsp40, to accelerate the ATPase activity [149]. In addition, Hsp40 scans for substrates, which it hands over to Hsp70 for refolding [150,151]. The interaction of Hsp70 with nucleotide and its ATPase function makes it amenable to drug targeting using ATP mimetics [152]. In the ATP-bound state, the Hsp70_NBD_ docks onto the Hsp70_SBD_ [153]. The allosteric function of Hsp70 is facilitated by its linker, and hence, the NBD:SBD interface of Hsp70 may constitute a promising target against the protein’s function. The design of prospective Hsp70 inhibitors against protozoan species has been largely inspired by the development of anticancer drugs targeting Hsp70. Several potential Hsp70 inhibitors have been identified, including adenosine analogues, malongonenes, pyrimidinones, fatty acids, and peptides (Figure 8).

### P. falciparum Hsp70 as a Drug Target

There are six Hsp70 members in *P. falciparum*. Four of these are canonical Hsp70s that localize to distinct cellular compartments: PfHsp70-1 (PF3D7_0818900; cytosol-localized), PfHsp70-2 (PF3D7_0917900; endoplasmic reticulum), PfHsp70-3 (PF3D7_1134000; mitochondrium), and PfHsp70-x (PF3D7_0831700), which is exported to the iRBC [1,110,154] The parasite also expresses two Hsp110 isoforms: PfHsp70-y (PF3D7_1344200; endoplasmic reticulum resident) and PfHsp70-z (PF3D7_0708800; cytosolic) [154,155]. Hsp110 proteins are Hsp70-like proteins that are known to function as nucleotide exchange factors (NEFs) of the canonical Hsp70 [156]. However, Hsp110 proteins possess independent chaperone function [157,158]. In addition, Hsp110 inhibits aggregation of its substrate in vitro in a nucleotide-independent fashion, as opposed to canonical Hsp70, whose capability to suppress proteins aggregation is inhibited by ATP [158,159].

PfHsp70-1 plays a central role in maintaining parasite proteostasis during stressful conditions [159,160]. PfHsp70-1 is stress-inducible and its expression at the RBC stage of the parasite life-cycle is thought to be important for both parasite cyto-protection and pathogenicity of malaria [142,161]. PfHsp70-1 is known to function as a chaperone responsible for thermo-tolerance [160,162,163]. As such, it is a prominent target in the development of small molecule inhibitors against the parasite. Cockburn and colleagues [164] identified novel PfHsp70-1-inhibiting compounds extracted from *Leptogorgia gilchristi*, a Mozambique sea fan. The malonganenone derivatives (A, B, and C) selectively inhibited the ATPase activity of both cytosolic and RBC exported *P. falciparum* Hsp70 (PfHsp70-x), but not the human Hsp70 (HSPA1A) (Figure 8) [152,164]. The same study also identified lapachol, a 1.4 naphthoquinone extract that selectively inhibited the ATPase activities of both parasite Hsp70s (PfHsp70-1 and PfHsp70-x), but not that of human Hsp70 (Figure 8) [154,164]. The functional assays conducted to validate sensitivity of Hsp70 were limited to ATPase assays, as heat aggregation suppression function of human HSPA1A could not be ascertained as this protein was not stable at 48 °C (the temperature at which the assay was conducted) [164]. In addition, another study using *P. falciparum* (Clones W2 and D6) cells maintained at the red blood stages showed interesting prospects, as some aminoquinones had higher potency than chloroquine against the W2 clone [165]. The findings based on malonganenone compounds as a potential antimalarial drug serve as a platform for further development of synthetic scaffolds of malongonenone backbone.

Our group previously showed that polymyxin (PMB), a cyclic peptide, and (-)-Epigallocatechin-3-gallate (EGCG), a green tea flavonoid, are both capable of binding to the NBD of PfHsp70-1 in micromolar range [166,167]. In addition, the two compounds also inhibited the chaperone and ATPase activities of PfHsp70-1 and PfHsp70-z [166,167]. Using cell culture-based studies, it was observed that EGCG was effective at inhibiting parasite growth with a modest IC_50_ of 2.9 µM [167]. On the other hand, PMB was not effective against parasite growth, and this could be because the peptide is restricted to the cell membrane, as it is known to bind to lipopolysaccharides [166]. However, efforts at repurposing approved drugs is an important part in the development of antiparasitic drugs.

PfHsp70-2, also known as *P. falciparum* immunoglobulin heavy chain binding protein (PfBiP/PfGrp78), is an ER-localized member whose expression at the RBC stage of the parasite life cycle is stress-inducible [168]. PfHsp70-2 facilitates the import of proteins into the ER and to ensure they are folded properly [169]. PfHsp70-2 is thought to be an essential molecule whose inhibition results in parasite death [170]. The repurposed anticancer drugs Apoptozole, MKT-077, and VER-15008 exhibited significant binding affinities for PfHsp70-2 and further demonstrated potent antiplasmodial activity to both *P. falciparum* 3D7 and W2 strains (Figure 8) [170]. It is, however, possible that the observed antiplasmodial activities of MKT-077 and VER-15008 may be attributed to their possible inhibition of pathways other than the PfHsp70-2. In cancer-based studies, MKT-077 was further modified to YM-08 with increased solubility to cross the plasma membranes [171]. This shows that more effort needs to be devoted towards modifying compounds known to inhibit parasite growth in order to improve their efficacy.

PfHsp70-x shares high amino acid sequence identity with PfHsp70-1 and localizes in the PV, and is also exported to the infected RBC cytosol (Figure 5) [110]. PfHsp70-x is thought to facilitate the refolding and activation of exported parasite proteins [172,173]. Unexpectedly, studies have shown that PfHsp70-x is dispensable for parasite survival and export of proteins of parasitic origin to the RBC cytosol [174,175]. However, its role may be important in the export of cargo proteins that are not required during the RBC stage of the parasite, such as cytoadherence proteins [107]; hence, PfHsp70-x may still be essential for clinical malaria development. In addition, an in vitro study demonstrated that PfHsp70-x binds to human Hop [176]. The absence of PfHop in the iRBC suggests that the EEVN motif of PfHsp70-x may modulate the possible interaction of PfHsp70-x with human Hop and human Hsp90. This in turn suggests that PfHsp70-x may cooperate with human Hsp90 to form an inter-species protein folding system in the RBC.

## 6. Hsp40 Family of Molecular Chaperones

Hsp40s (known as DnaJ in prokaryotes) are characterized by the presence of signature J-domain, made up of around 70 amino acids, on average [177]. There are four types (I–IV) of Hsp40s, mainly distinguished based on the presence of functional domains and motifs, which include a J-domain; presence of His, Pro, and Asp (HPD) motif; the glycine–phenylalanine (GF)-rich region and cysteine-rich domain (CRR) [44,178]. Type I Hsp40s are composed of all the typical *E. coli* DnaJ domain architecture: A J-domain (with HPD motif), GF-rich region, and CRR [43]. Type II Hsp40s are similar to Type I Hsp40s, but lack the CRR [43]. Hsp40s of type III harbor the typical J-domain, but lack both the GF region and the CRR. Type IV Hsp40s possess an atypical J-domain (without the HPD motif), which is variably located on the protein, and like the type IIIs, they lack the GF region and the CRR [43].

Hsp40s recruit substrates for Hsp70 [179]. Besides this, they stimulate the Hsp70 ATPase activity [180,181,182]. However, some type III and IV Hsp40s are incapable of modulating the ATPase activity of Hsp70 [43], and their function remains largely elusive.

### 6.1. Hsp70–Hsp40 Partnership as a Drug Target

Hsp40 proteins are known to bind substrate first, which they then pass to Hsp70 [181]. Hsp40s act in time and space to regulate function of their respective Hsp70 partners. For example, Hsp40s localize at the ribosomal exit tunnel, from where they recruit client polypeptides and deliver them to Hsp70 for folding [183]. In this case, they act as “targeting factors” for Hsp70 clients, as they thus recruit the latter to sites where their clients are located (Figure 1) [181,183]. Since Hsp70 depends on Hsp40 for recruitment to the site of action and for delivery of substrates and modulation of its ATPase activity, Hsp40s regulate the functional specificity of Hsp70. The number of Hsp40s varies in different organisms. Prokaryotes possess fewer Hsp40 members (*E. coli*, 6; *Synechococchus elongates*, 3; [182,183]) than their eukaryotic counterparts. For example, humans have at least 50 Hsp40s; at least 49 in *P. falciparum* [44,183], > 60 in *Leishmania* spp., and 33 in *T. gondii* [1]. The varied distribution of Hsp40 members across species is thought to influence the functional specificity of Hsp70s [151].

### 6.2. Targeting the P. falciparum Hsp70 and Hsp40 Partnership

Of the 49 Hsp40s in *P. falciparum,* 36 members (2 type I Hsp40s, 8 type II, and 26 type III) are thought to directly interact with Hsp70s through the HPD motif on their respective J-domains [44]. On the other hand, type IV Hsp40s (13 in total) lack a conserved HPD motif, and as such, their role remains to be fully established [44]. Nineteen Hsp40s possess the PEXEL signal sequence for export to the iRBC cytosol, where they are thought to facilitate recruitment of proteins of parasitic origin towards refolding chaperone systems, and subsequent host cell remodeling through interaction with host cell cytoskeleton [44]. Remodeling of the infected RBC characterizes malaria pathology, and as such, Hsp40s play an important role in parasite survival and pathogenesis. For example, PF11_0034 (PF3D7_1102200), RESA (PF3D7_0102200), and PF10_0382 (PF3D7_1039100) have been shown to be essential for parasite pathogenesis [44,184], making them druggable candidates. The distinct species-specific Hsp70–Hsp40 partnerships that owe themselves to the varied complement of Hsp40 members in each species give rise to specialized chaperone functions across species [151]. This suggests that *P. falciparum* Hsp40–Hsp70 functional cooperation constitutes a selectable target in antimalarial drug design, despite the general conservation of this pathway [185].

In *P. falciparum*, type I and type II Hsp40s functionally associate with Hsp70 to activate the latter’s ATPase activity [186]. Their functional partnership with Hsp70 is defined by substrate recruitment and subsequent activation of the otherwise rate-limiting ATP hydrolysis by Hsp70. The role of Hsp40s as regulators of ATPase function of Hsp70 is more amenable to possible drug targeting [185]. The interaction between Hsp70 and Hsp40 is mediated through the J-domain of the Hsp40 [187]. This suggests that targeting the J-domain–Hsp70 interaction presents a bottleneck that disrupts this functional complex, arresting the folding of Hsp70-dependent client proteins.

Type I *P. falciparum* Hsp40 (PfHsp40; PF3D7_1437900) interacts with PfHsp70-1, forming a PfHsp70-1–PfHsp40 functional complex [43]. Using comparative functional assays, selective inhibition of PfHsp40 (PF3D7_1437900)-stimulated ATPase activity of PfHsp70-1 by pyrimidone peptoides DMT002264 and MAL3-39 over human orthologues was observed [178]. Subsequent to that, malanganenone A, B, C, and lapachol were also reported to inhibit the Hsp40-stimulated ATPase activity of *P. falciparum* Hsp70 [152,164].

The malaria parasite mitochondrium-localized PfHsp70-3 [188] has been implicated in facilitating the translocation of proteins across the two mitochondrial membranes [169,189]. Most of the proteins that function in the mitochondrial matrix are nuclear encoded, as the mitochondrial genome is reported to encode for only two functional proteins [190]. Thus, nuclear encoded proteins are thought to be translocated across the two mitochondrial membranes with the assistance of PfHsp70-1 on the cytosol side, while PfHsp70-3 receives the peptides on the mitochondrion matrix [169]. Given these key functions of Hsp70 of malaria parasite make essential for parasite development, and may thus present a chink in the armor of the parasite with respect to targeted inhibition. However, it should be noted that a small molecule inhibitor targeting PfHsp70-3 would have to be soluble enough to penetrate the double layer of membranes to reach the mitochondria.

## 7. Hsp110 Family of Proteins

Hsp110s are distant family members of the Hsp70 superfamily [191]. Hsp110s are localized in the cytosol and the ER-localized members are Grp170 [192]. Hsp110s are distinct from the archetypical Hsp70 on account of their possession of a much larger SBD due to the presence of a long acidic loop in this domain [193,194]. Consequently, Hsp110s exhibit holdase (protein aggregation suppression) function and lack the capability to refold substrates, as they lack inter-domain allosteric regulation [195]. These proteins function as nucleotide exchange factors of canonical Hsp70.

### 7.1. Interaction of Hsp70 with Hsp110

Nucleotide exchange factors (NEFs) indirectly modulate substrate dwell time on Hsp70. NEFs acting as substrate release factors improve the Hsp70 operational efficiency [156,196]. Several NEFs employed by Hsp70 are divergent and unrelated, but serve the same role. For nucleotide exchange, prokaryotes (and mitochondria) harbor only the GroP-like gene E (GrpE) [197]. On the other hand, eukaryotes possess several NEFs that can be grouped into three main classes: Bcl2-associated athanagene (BAG-1-3) [198], heat shock protein binding protein 1 (HspBP1) [199], and Hsp110 [156]. The unique NEF compliment in various organisms is an attractive feature for their selective targeting.

### 7.2. Targeting the Hsp110–Hsp70 Interaction

The NEFs of PfHsp70-1 are yet to be fully established. Nucleotide exchange function of Hsp70 defines substrate dwell time on the chaperone, and this in turn influences the fate of substrates, as those proteins that are not folded are channeled towards degradation [200]. The nucleotide exchange cycle of Hsp70 is a promising druggable pathway in *P. falciparum* [53]. However, to design selective inhibitors targeting this pathway, there is need to fully map out the protein players involved in the process. A cytosol-localized Hsp110 family member, PfHsp70-z, is the only proposed NEF of PfHsp70-1 [154,169]. In addition, PfHsp70-z, is essential for parasite survival [201]. This suggests that PfHsp70-z could be the sole NEF that modulates the function of cytosolic PfHsp70-1, and its inhibition would lead to parasite death [158].

The Hsp110 protein family members are thought to facilitate nucleotide exchange activity of Hsp70 [156]. PfHsp70-y, an ER-localized Grp170 is likely to serve as the NEF for the ER-localized PfHsp70-2 [169]. The ER requires a robust NEF to modulate Hsp70 protein refolding due to the abundance of products of oxidative stress that accumulate on account of both the production of hydrogen peroxide and antimalarial drug pressure [202,203]. The independent chaperone function of PfHsp70-z has been demonstrated in vitro, suggesting that apart from its possible role as an NEF, it has a direct role on parasite proteostasis [157,158,166,167]. PfHsp70-z was shown to interact with PfHsp70-1 in a nucleotide-dependent manner, and ATP enhances the association of the two molecules [158]. The nucleotide-dependent interaction of the two chaperones heightens its proposed NEF role [157,204]. Furthermore, our group previously observed that PMB and EGCG inhibit the interaction of PfHsp70-1 with its functional partner PfHsp70-z [158,166,167]. Both PMB and EGCG bound to both PfHsp70-1 and PfHsp70-z to induce conformational changes, leading to abrogation of the interaction of the two proteins [158]. This suggests that Hsp70 inhibitors go beyond inhibiting their chaperone function to further abrogate their functional network.

## 8. Conclusions and Future Perspectives

The merit of targeting heat shock proteins lies in the role they play in parasite proteostasis. They are involved in proper folding of proteins that carry out diverse functions in the parasite. Not surprisingly, several parasite heat shock proteins are essential for survival and cell stage differentiation. The main classes of heat shock proteins function in networks, and as such, their expression is coordinated. This in turn explains their capability to functionally compensate for one another. For example, inhibition of Hsp90 was observed to stimulate the heat shock response characterized by upregulated expression of Hsp70 as a compensatory mechanism in cancer cells [205]. In addition, some parasites harbor an expanded Hsp40 complement. This results in functional redundancy [206]. For these reasons, it seems the most prospective approach to target heat shock proteins against parasitic infections remains to inhibit them in combination with drugs targeting other molecules. In this respect, targeting parasite heat shock proteins appears most promising as a means to recover the action of previously effective drugs and currently effective drugs as the parasites eventually master how to evade them.

## Figures and Tables

**Figure 1 ijms-20-05930-f001:**
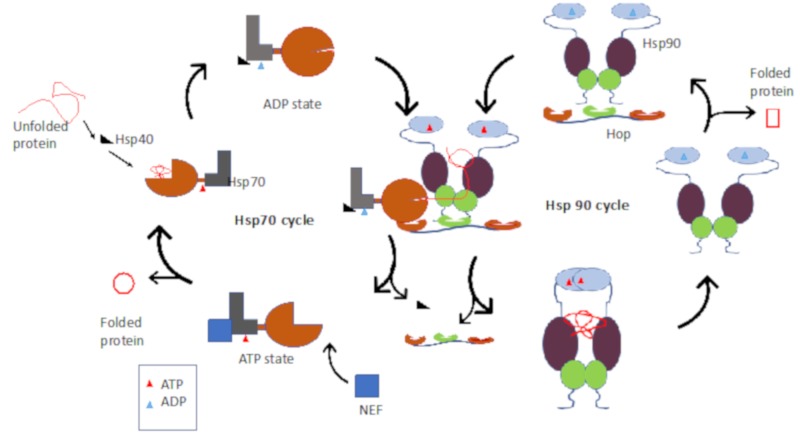
Hsp90–Hsp70 chaperone pathway. Chaperone-mediated substrate folding by the Hsp70–Hsp40–nucleotide exchange chaperone complex in cooperation with Hsp90 via Hop mediation.

**Figure 2 ijms-20-05930-f002:**
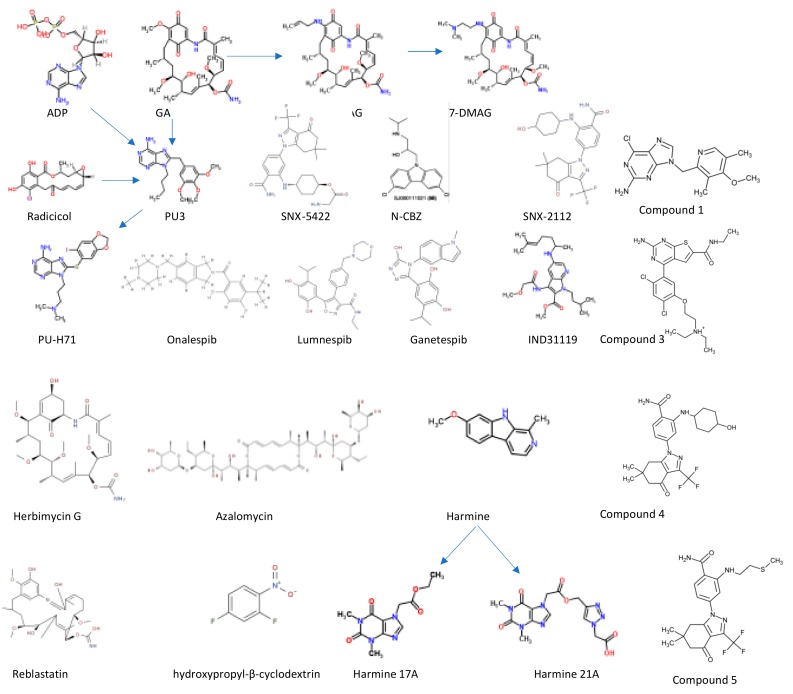
The structure of common Hsp90 inhibitors. Geldanamycin (GA), the first identified Hsp90 inhibitor, has been used for further development of 17-AAG and 17-DMAG. The crystal structure of Hsp90 NTD in complex with ADP/GA and radicicol were used for development of purine scaffold from PU3 to generate Debio 0932, PU-H71, MPC-3100, and BIIB021(compound **1**), among others [51,62]. Other compounds such as onalespib, luminesbip, ganetespib, IND31119, herbimycin, and azalomycin were also identified as candidate Hsp90 inhibitors [63]. Harmine and its derivatives 17A and 21A are also promising Hsp90 inhibitors [64].

**Figure 3 ijms-20-05930-f003:**
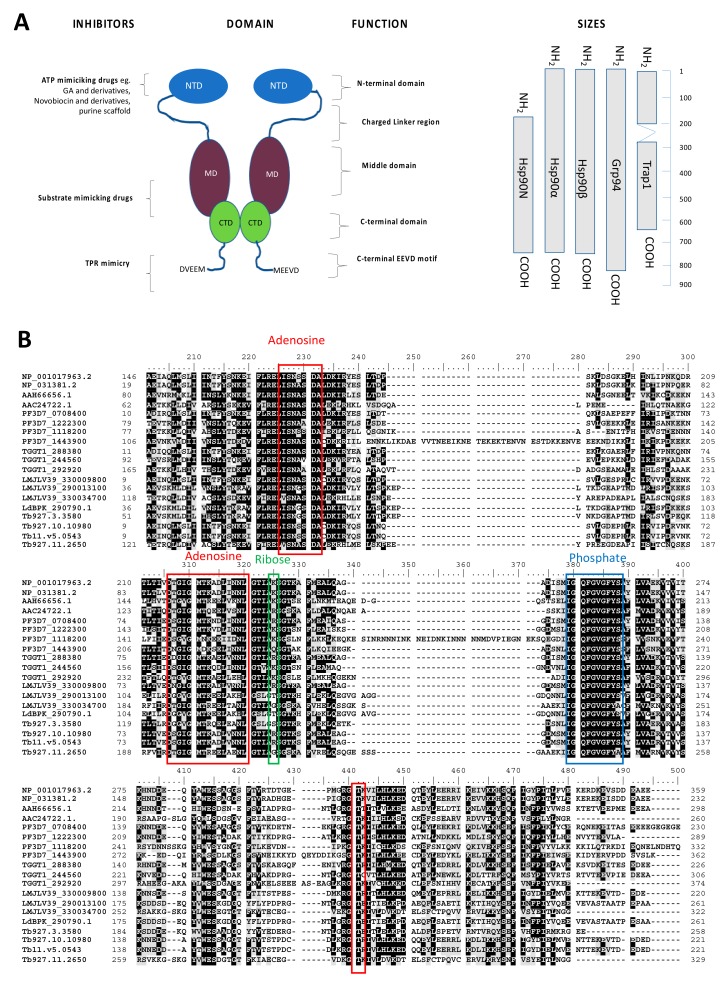
Hsp90 domain organization and mode of action of the inhibitors targeting it. (**A**) The schematic shows Hsp90 domain organization across the various isoforms of the protein. (**B**) Multiple sequence alignment of the nucleotide binding domain of Hsp90, and highlighted are residues that bind adenosine (red box), ribose (green box), and phosphate (blue box). The multiple sequence alignments of the nucleotide binding domains of the Hsp90 homologues and orthologues from *Homo sapiens* were retrieved from https://www.ncbi.nlm.nih.gov/protein (accession # NP_001017963.2, NP_031381.2; AAH66656.1, AAC24722.1); *P. falciparum* from www.plasmoDB.org (accession # PF3D7_0708400, PF3D7_1222300, PF3D7_1118200, PF3D7_1443900); *T.gondii* from www.toxoDB.org (accession # TGGT1_288380, TGGT1_244560, TGGT1_292920); *L. major* from www.TriTrypDB.org (accession # LMJLV39_330009800, LMJLV39_290013100, LMJLV39_330034700); *Leishmania donovani* from www.TriTrypDB.org (accession # LdBPK_290790.1); and *T. brucei* (T accession # Tb927.3.3580; Tb927.10.10980, Tb11.v5.0543; Tb927.11.2650).

**Figure 4 ijms-20-05930-f004:**
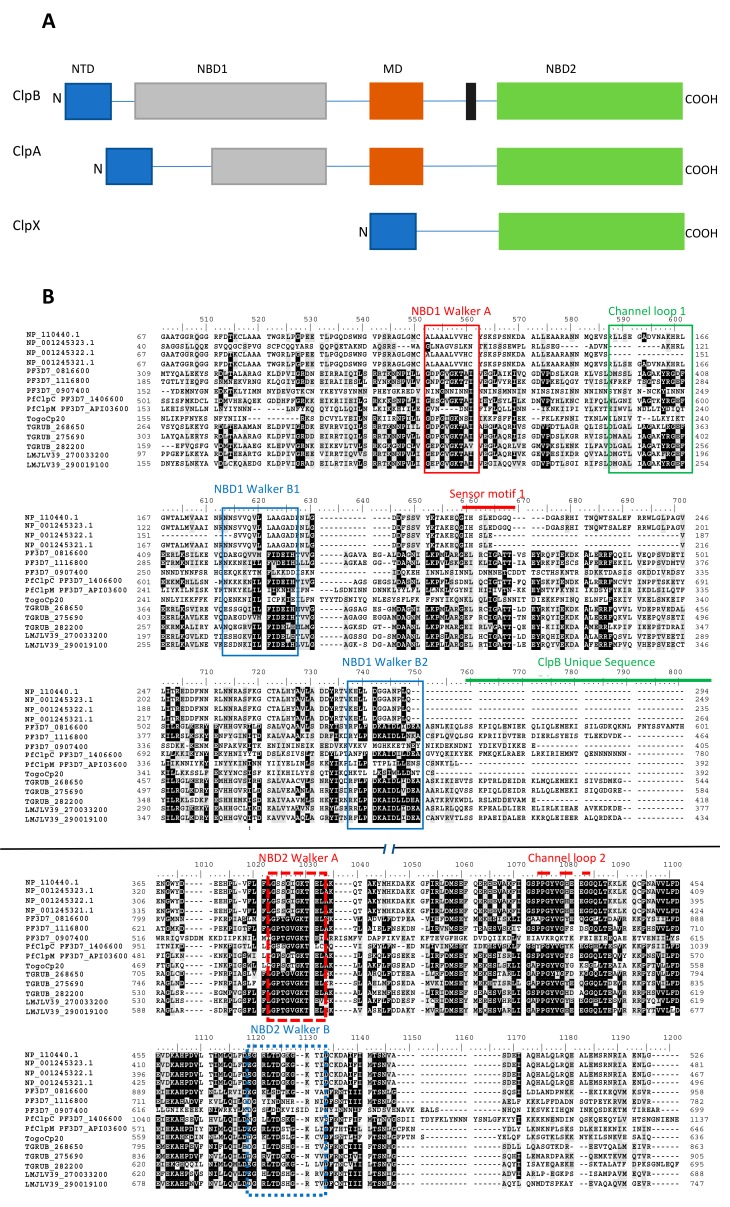
The schematic representing Hsp100 domain organization and sequence alignment. (**A**) Hsp100 isoforms share domain architecture. Some of their members include Hsp104/CpC comprised of an N-terminal substrate binding domain (NBD), two nucleotide binding domains, NBD1/2, and a middle domain (MD). Another Hsp100 member, ClpA, lacks the middle domain, while ClpX possesses a single NBD. (**B**) The multiple sequence alignments of the nucleotide binding domains of the Hsp100 were conducted using the sequences retrieved as follows: *Homo sapiens,*
https://www.ncbi.nlm.nih.gov/protein (accession # NP_1104400.1, NP_001245323.1, NP_001245322.1, NP_001245321.1); *P. falciparum* from www.PlasmoDB.org (accession # PF3D7_0816600, PF3D7_1116800, PF3D7_0907400, PF3D7_1406600, PF3D7_API03600); *T. gondii* retrieved from www.ToxoDB.org (accession # Togocp20, TGRUB_268650, TGRUB_275690, TGRUB_2822200); and *L. major* retrieved from www.TriTrypDB.org (accession # LMJV39_270033200, LMJLV39_290019100).

**Figure 5 ijms-20-05930-f005:**
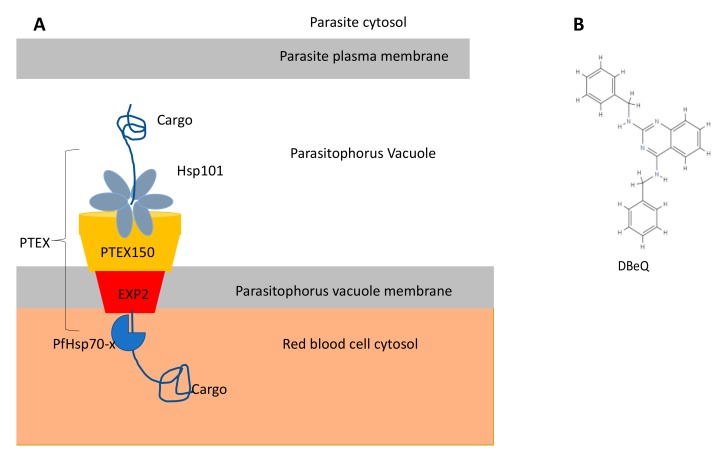
The *Plasmodium* translocon of exported proteins (PTEX) system and the chemical structure of Hsp100 inhibitor. (**A**) The schematic shows the parasite and infected human red blood cell (RBC) interface and the architecture of the PTEX core. The protein cargo (exported protein) is first unfolded by the Hsp101 in the parasitophorus vacuole and is delivered to the PTEX150, which threads it towards the RBC cytosol through EXP2. Another exported chaperone, PfHsp70-x, present in the parasitophorous vacuole (PV) and the cytosol of the RBC is implicated in the trafficking of exported parasite proteins and their subsequent refolding on delivery [110,111]. (**B**) Chemical structure of DBeQ, a known ATP competitive inhibitor [91].

**Figure 6 ijms-20-05930-f006:**
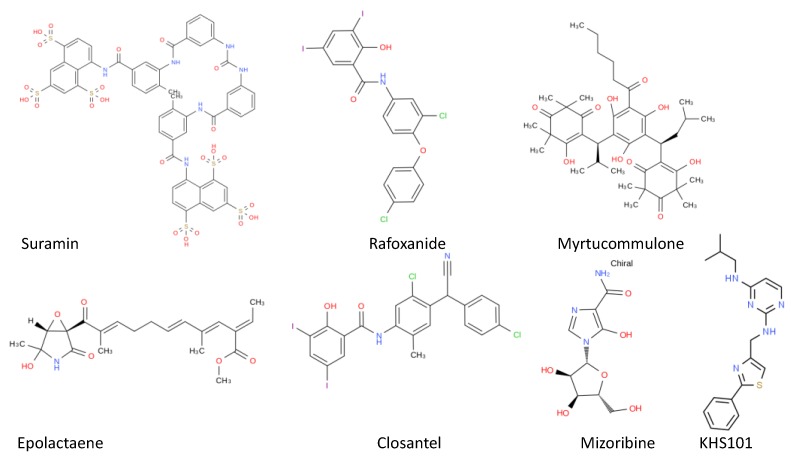
Candidate compounds that target Hsp60. Various chemical structures representing potential Hsp60 inhibitors that target both the ATPase and substrate binding activities of the chaperonin are illustrated.

**Figure 7 ijms-20-05930-f007:**
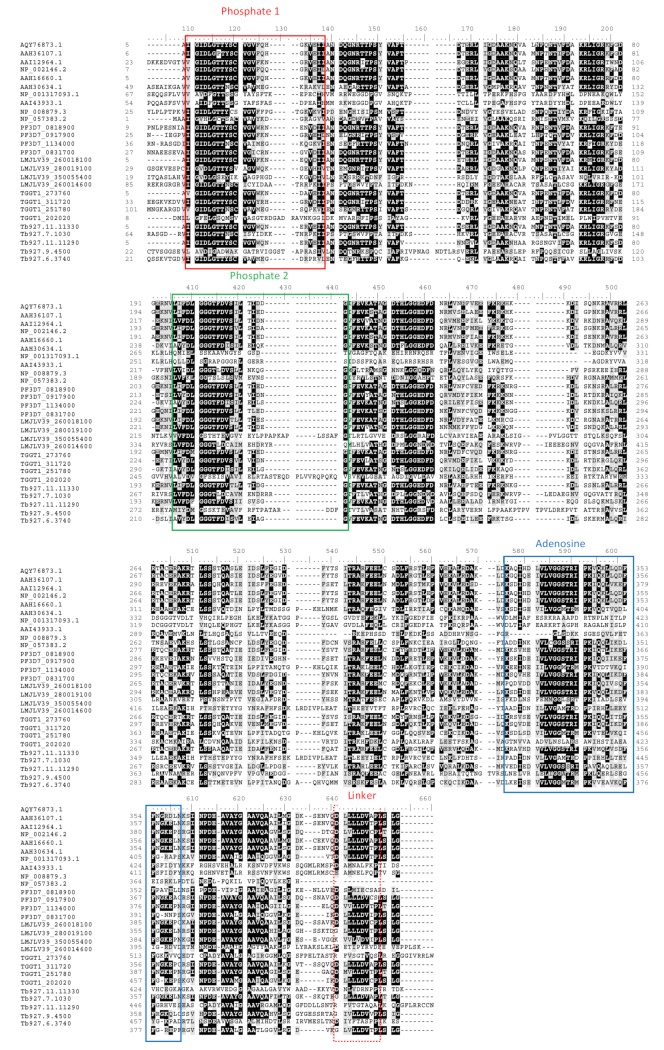
Multiple sequence alignment of Hsp70 homologues. The nucleotide binding domain of Hsp70 (nucleotide binding motifs are highlighted with colored boxes). Hsp70 amino acid sequences of various species origin were retrieved as follows: *Homo sapiens,*
https://www.ncbi.nlm.nih.gov/protein (accession # AQY7673.1, AAH36107.1, AAI1296.1, NP_002146.2, AAH1660.1, AAH30634.1, NP_001317093.1, AA143933.1, NP_008879.3, NP_057383.2); *P. falciparum* Hsp70s from www.PlasmoDB.org (accession # PF3D7_0818900, PF3D7_0917900, PF3D7_1134000, PF3D7_0831700); *L. major* from the www.TriTrypsDB.org (accession # LMJLV39_26008100, LMJLV39_280019100, LMJLV39_350055400, LMJLV_260014600); *T. gondii* from the www.ToxoDB.org (accession #TGGT1_273760, TGGT1_311720, TGGT1_251780, TGGT1_202020); and *T. brucei* from www.TrytripsDB.org (accession # Tb927.11.11330, Tb927.7.1030, Tb27.11.11290, Tb27.9.4500, Tb927.6.3740).

**Figure 8 ijms-20-05930-f008:**
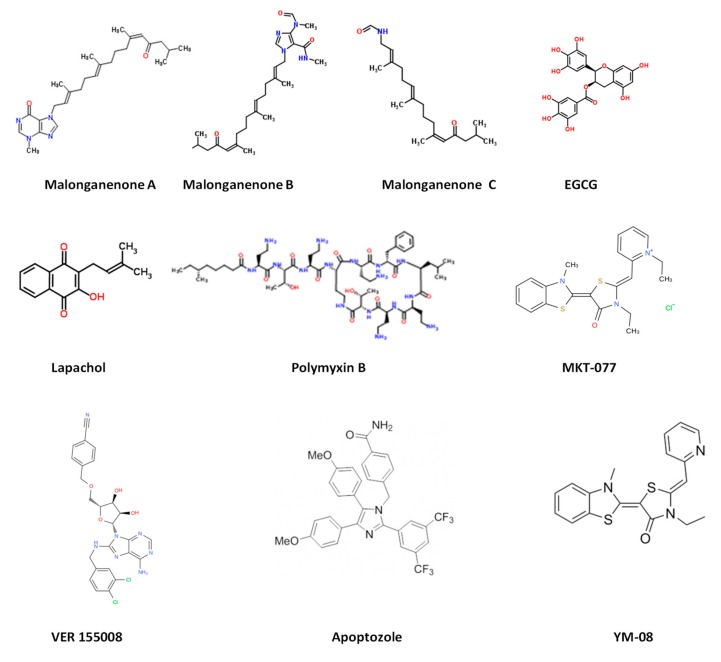
Chemical structures of Hsp70 inhibitors.

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
