# Peer review of "Small Molecule Inhibitors Targeting the Heat Shock Protein System of Human Obligate Protozoan Parasites"

_ijms, 2019, doi:10.3390/ijms20235930_

Round 1
Reviewer 1 Report
Zininga and Shonhai have prepared a comprehensive overview of possible chaperone targets for the treatment of parasitic infection. They first introduce the basic concepts of how parasites use chaperones and then arrange the manuscript according to structural classes (e.g. Hsp90, Hsp70, Hsp104, etc). In each section, they succinctly summarize the biochemistry and then discuss known inhibitors, with a focus on those that have been shown to have anti-parasitic activity. While the review will likely be a good place for young scientists to start learning about the field, there are some mistakes and missed opportunities that somewhat dampen enthusiasm. A few suggestions:
As the authors rightly state, a major problem with targeting chaperones as drug targets is their high conservation in all kingdoms of life. It seems that any chaperone-targeted drug will have significant toxicity unless it is carefully crafted to avoid binding the human equivalent. Indeed, the pivotal nature of this issue warrants more focus throughout the manuscript. For example, a structural and/or sequence comparison of human and parasite Hsp90, Hsp70, Hsp60, etc. would be an excellent addition – especially if the focus was placed on potentially druggable sites, such as ATP-binding pockets. Such structural comparisons would make the review more scholarly and impactful. Further, the authors could comment on which class of chaperones might be more/less amenable to selective targeting. The authors are confused about the status of Hsp90 inhibitors. On line 209, they erroneously claim that efforts to target the NBD have not been successful – while on Line 149-150, they claim that efforts to target the EEVD have been prominent. This is the opposite of the truth. All of the molecules that have advanced to clinic as Hsp90 inhibitors target the NBD. Efforts to target the CTD are more nascent. It should be made more clear that mammals lack the Hsp104 class of disaggregating chaperones. This could be an important point for selective targeting of parasites. Figure 6. An exciting, new Hsp60 inhibitor, KHS-101, is not included (Polson et al. 2018 Sci. Transl. Res.). The rest of the molecules in this category are quite reactive and/or promiscuous and not “drug-like”. Line 473. The Hsp40 class of chaperones has been re-named as J-domain proteins (JDPs), see Kampinga et al. 2018 Cell Stress Chaperones. This renaming is important because only a portion of the family members are of 40 kDa. In addition, the JDPs that lack the invariant HPD motif (termed type IV in the current review) are no longer considered to be part of the family, as they cannot interact with Hsp70s. Finally, many type III (typically called DnaJC family members) do, in fact, stimulate ATPase activity, so the statement on line 487 is false.
Minor:
Lines 117-125, the major ideas are duplicated from the previous paragraph. These are redundant and can be removed. Line 96. Only the cytoplasmic forms of the Hsp70 and Hsp90 proteins contain an EEVD motif, while the ER and mitochondrial forms do not. Care should be taken here and elsewhere in the manuscript about over-generalizing. Line 392. While the ATPase activity of prokaryotic Hsp70s is stimulated by peptide/protein substrates, the mammalian systems are not (see recent work by Gierasch and others). Line 164. What is the evidence that Hsp90 inhibitors trap the ADP-bound state? This conclusion is often presumed in the literature, but with little/no direct evidence. Line 132. Remove “in addition the basal.”
Reviewer 2 Report
Zininga and Shonhai have done an impressively thorough job at summarizing a huge number of efforts to characterize and to target molecular chaperones of protozoan parasites. It provides a comprehensive and relatively well readable overview of what's been done and what may still need to be done to develop promising new drugs. Although I list a large number of specific issues in the following, none is really fundamental and they should all be easy to deal with.
Specific comments:
Selective choice of only some human parasites: although the Abstract specifically mentions the kinds of parasites that will be discussed, perhaps the title could be more specific by including the statement "protozoan parasites". There are obviously lots of other parasites that this review does not discuss. Line 46: "arthopod", not anthropod. Page 2, from line 66: the concept of "genetic erosion" is erroneously attributed to ref. 9. It was proposed in ref. 10, but for an entirely different pathogen. The argument "genetic erosion" for a robust protein quality system in protozoan parasites seems misplaced. Page 2, from line 65: an argument that is strangely missing (for a robust proteostasis system) is that parasites alternate between hosts/vectors/environments with very different working temperatures. Page 2, line 93: one can definitely do better than ref. 19 with recent reviews, for example the recent review by the Buchner group. Moreover, references 20 and 21 seem to be somewhat random, too. Note that the C-terminal sequence in Hsp90 is MEEVD, which includes but is distinct from EEVD (as in Hsp70). In fact, in some of the parasites discussed here, it is MEQVD. Line 120: "replaces ADP by ADP"? Figure 1: It is not clear (nor necessary) why PP5 is singled out as a TPR co-chaperone. Line 132: the statement "In addition..." is truncated. Page 4, upper paragraph: the speculation that high ATPase activity in any way correlates with inhibitor sensitivity or with functional requirements should be dealt with more carefully. The Buchner lab has shown (Zierer et al., 2016) that ATPase activity and in vivo activity do not correlate. What this all means remains to be clarified. Line 149: while it is fine to discuss this class of inhibitors, they can hardly be called "prominent". Line 184: what is "ganetespib IND31119"? Chapter 2.5: the authors overlooked that Pf has an additional Hsp90 isoform for the apicomplexan-specific apicoplast (PF3D7_1443900). Strangely, this is mentioned as "a truncated Hsp90 with a missing cytosolic signal". However, it is not only not truncated (rather, it has extensions on both ends), but it is also totally unclear what a "cytosolic signal" should be and what Fig. 3 would say about that. Line 196: "selectiviting"? Ref. 67: This is not the correct Wang et al. Line 213: The reference to the work of Wang and colleagues does not have the proper citation format. Chapter 2.8: there is a nice paper by the Pizarro group (Pizarro et al., 2013, PLoS Neglect Trop Dis), which needs to be mentioned here. Lines 304 to 307: two incomplete sentences. Lines 381-383: these first few lines should be carefully edited. PfHsp70s: I don't see a difference in cellular compartments for PfHsp70-1 and PfHsp80-z. Hence, their presentation on page 12 and the statement of line 443 ("cytosolic counterpart") are weird. Hsp110 proteins: it is a bit confusing that they appear again in chapter 7 after having been mentioned/discussed in chapter 5. This could be sorted out more rationally, possibly also by referring to each other with minimal redundancy.
Author Response
We have addressed comments from reviewer 2 (see attached document)
Zininga and Shonhai have done an impressively thorough job at summarizing a huge number of efforts to characterize and to target molecular chaperones of protozoan parasites. It provides a comprehensive and relatively well readable overview of what's been done and what may still need to be done to develop promising new drugs. Although I list a large number of specific issues in the following, none is really fundamental and they should all be easy to deal with.
Specific comments:
Selective choice of only some human parasites: although the Abstract specifically mentions the kinds of parasites that will be discussed, perhaps the title could be more specific by including the statement "protozoan parasites". There are obviously lots of other parasites that this review does not discuss.
Author’s response: We have added the word ‘protozoan’ parasites to the title.
Line 46: "arthopod", not anthropod.
Author’s response: We have corrected and replaced ‘anthropod’ with ‘arthopod’.
Page 2, from line 66: the concept of "genetic erosion" is erroneously attributed to ref. 9. It was proposed in ref. 10, but for an entirely different pathogen.
Author’s response: We have corrected the referencing.
The argument "genetic erosion" for a robust protein quality system in protozoan parasites seems misplaced.
Author’s response: We appreciate the reviewer’s comments and we have removed the sentence on genetic erosion and replaced it with ‘For parasites to survive their proteome needs to be adept at meeting the demands of the hostile conditions prevailing within the alternating host/vectors environments characterized by variable physiological conditions such as pH, temperature and nutrient supply. Unlike free-living parasites, obligate parasites undergo extensive molecular evolution during their stint in the host [9]. This promotes production of mutated proteins, making their proteome generally aberrant’.
Page 2, from line 65: an argument that is strangely missing (for a robust proteostasis system) is that parasites alternate between hosts/vectors/environments with very different working temperatures.
Author’s response: We appreciate the reviewer’s comments and we have included the suggestion.
Page 2, line 93: one can definitely do better than ref. 19 with recent reviews, for example the recent review by the Buchner group.
Author’s response: We appreciate the reviewer’s suggestion, we have replaced our reference with ‘ 19.Biebl, M. M.; Buchner, J. Structure, function, and regulation of the Hsp90 machinery. CSH Perspect Biol, 2019, a034017’.
Moreover, references 20 and 21 seem to be somewhat random, too. Note that the C-terminal sequence in Hsp90 is MEEVD, which includes but is distinct from EEVD (as in Hsp70). In fact, in some of the parasites discussed here, it is MEQVD.
Author’s response: We appreciate the reviewer’s comments and we have included the suggestion.
Line 120: "replaces ADP by ADP"?
Author’s response: We have corrected the sentence to: ‘ADP is replaced by ATP’.
Figure 1: It is not clear (nor necessary) why PP5 is singled out as a TPR co-chaperone.
Author’s response: We have corrected the figure and removed PP5 from both the figure and figure legend.
Line 132: the statement "In addition..." is truncated.
Author’s response: We have corrected the sentence.
Page 4, upper paragraph: the speculation that high ATPase activity in any way correlates with inhibitor sensitivity or with functional requirements should be dealt with more carefully. The Buchner lab has shown (Zierer et al., 2016) that ATPase activity and in vivo activity do not correlate. What this all means remains to be clarified.
Author’s response: We have toned down the correlation on activity and function. The sentence has been reworded to: ‘The high basal ATPase activity of parasite Hsp90 such as the P. falciparum cytosolic homologue, PfHsp90 is more sensitive to inhibition than its human homologue’.
Line 149: while it is fine to discuss this class of inhibitors, they can hardly be called "prominent".
Author’s response: We have corrected the sentence by deleting the word ‘prominent’.
Line 184: what is "ganetespib IND31119"?
Author’s response: We have corrected the sentence by specifying that ganetespib is also named STA-9090 as in line 198, and we have also separated ganetespib from IND31119 by inserting a comma.
Chapter 2.5: the authors overlooked that Pf has an additional Hsp90 isoform for the apicomplexan-specific apicoplast (PF3D7_1443900). Strangely, this is mentioned as "a truncated Hsp90 with a missing cytosolic signal". However, it is not only not truncated (rather, it has extensions on both ends), but it is also totally unclear what a "cytosolic signal" should be and what Fig. 3 would say about that.
Author’s response: We appreciate the reviewer’s comment and as such we have corrected the apicoplast localised Hsp90 and deleted the words ‘missing cytosolic signal’ which were misleading. In addition, we have included the molecule on the multiple sequence alignment in figure 3B.
Line 196: "selectiviting"?
Author’s response: We have corrected the word to ‘sensitivity’.
Ref. 67: This is not the correct Wang et al.
Author’s response: We have corrected the reference by inserting the following ref [67] ‘Wang, T.; Mäser, P.; Picard, D. Inhibition of Plasmodium falciparum Hsp90 contributes to the antimalarial activities of aminoalcohol-carbazoles. J. Med. Chem. 2016, 59, 6344-6352’.
Line 213: The reference to the work of Wang and colleagues does not have the proper citation format.
Author’s response: We appreciate the reviewer’s comments and we have corrected the reference to ‘Wang and colleagues [67]’
Chapter 2.8: there is a nice paper by the Pizarro group (Pizarro et al., 2013, PLoS Neglect Trop Dis), which needs to be mentioned here.
Author’s response: We have inserted the reference’ [87] Pizarro, J.C.; Hills, T.; Senisterra, G.; Wernimont, A.K.; Mackenzie, C.; Norcross, N.R.; Ferguson, M.A.; Wyatt, P.G.; Gilbert, I.H.; Hui, R. Exploring the Trypanosoma brucei Hsp83 potential as a target for structure guided drug design. PLoS Negl. Trop. Dis, 2013, 7, e2492’.
Lines 304 to 307: two incomplete sentences.
Author’s response: We have edited sentences.
Lines 381-383: these first few lines should be carefully edited.
Author’s response: We have edited lines 381-383 to read ‘Hsp70 is one of the most evolutionary conserved superfamily of Hsps that occur in all domains of life [140]. In archaea and eubacteria Hsp70s, are referred to as DnaK. Hsp70s are ubiquitous molecules, and some of their homologues are stress inducible’.
PfHsp70s: I don't see a difference in cellular compartments for PfHsp70-1 and PfHsp80-z. Hence, their presentation on page 12 and the statement of line 443 ("cytosolic counterpart") are weird.
Author’s response: We have removed the words ‘cytosolic counterpart’.
Hsp110 proteins: it is a bit confusing that they appear again in chapter 7 after having been mentioned/discussed in chapter 5. This could be sorted out more rationally, possibly also by referring to each other with minimal redundancy.
Author’s response: We have removed the redundancy in chapter 7 by rewording to: ‘Heat shock protein 110 (Hsp110) is a member of the Hsp70 super-family of molecular chaperones [196]. Hsp110s are localised in the cytosol while Grp170 occurs in ER [197]. Hsp110s have larger substrate binding domains through distinct insertion of a long acidic loop and have significant sequence divergence from the archetypical Hsp70 [198-199]. Consequently, Hsp110s have been found to exhibit chaperone function limited to holdase function (protein aggregation suppression) as they lack inter-domain allosteric regulation [200]. These proteins function as nucleotide exchange factors of canonical Hsp70s’.
Round 2
Reviewer 2 Report
The manuscript has been extensively edited and my comments have been properly addressed. One minor little thing that should be corrected before acceptance is Fig. 1. The version associated with the revised manuscript still mentions PP5 (middle right).
Author Response
Reviewer round 2 comments:
The manuscript has been extensively edited and my comments have been properly addressed. One minor little thing that should be corrected before acceptance is Fig. 1. The version associated with the revised manuscript still mentions PP5 (middle right).
Author Feedback: We are grateful to the reviewer for the feedback. In our previous revised version, we removed all references to PP5 in figure 1 (both in the image and figure legends). We hope that the editorial office is sending the correct updated MS version to the reviewers.
As we addressed this before, we have made no changes to the current version (on the journal portal that we access as authors).
Thank you, again.